# Assessment of the Impact of Deformable Registration of Diagnostic MRI to Planning CT on GTV Delineation for Radiotherapy for Oropharyngeal Carcinoma in Routine Clinical Practice

**DOI:** 10.3390/healthcare6040135

**Published:** 2018-11-24

**Authors:** Alice Taylor, Mehmet Sen, Robin J. D. Prestwich

**Affiliations:** 1School of Medicine, Worsley Building, University of Leeds, Leeds LS2 9JT, UK; um13a2t@leeds.ac.uk; 2Department of Clinical Oncology, St. James’s University Hospital, Leeds Cancer Centre, Beckett Street, Leeds LS9 7TF, UK; mehmet.sen@nhs.net

**Keywords:** radiotherapy, deformable image registration, magnetic resonance imaging, oropharyngeal squamous cell carcinoma, gross tumour volume

## Abstract

Background: Aim of study was to assess impact of deformable registration of diagnostic MRI to planning CT upon gross tumour volume (GTV) delineation of oropharyngeal carcinoma in routine practice. Methods: 22 consecutive patients with oropharyngeal squamous cell carcinoma treated with definitive (chemo)radiotherapy between 2015 and 2016, for whom primary GTV delineation had been performed by a single radiation oncologist using deformable registration of diagnostic MRI to planning CT, were identified. Separate GTVs were delineated as part of routine clinical practice (all diagnostic imaging available side-by-side for each delineation) using: CT (GTV_CT_), MRI (GTV_MR_), and CT and MRI (GTV_CTMR_). Volumetric and positional metric analyses were undertaken using contour comparison metrics (Dice conformity index, centre of gravity distance, mean distance to conformity). Results: Median GTV volumes were 13.7 cm^3^ (range 3.5–41.7), 15.9 cm^3^ (range 1.6–38.3), 19.9 cm^3^ (range 5.5–44.5) for GTV_CT_, GTV_MR_ and GTV_CTMR_ respectively. There was no significant difference in GTV_CT_ and GTV_MR_ volumes; GTV_CTMR_ was found to be significantly larger than both GTV_MR_ and GTV_CT_. Based on positional metrics, GTV_CT_ and GTV_MR_ were the least similar (mean Dice similarity coefficient (DSC) 0.71, 0.84, 0.82 for GTV_CT_–GTV_MR_, GTV_CTMR_–GTV_CT_ and GTV_CTMR_–GTV_MR_ respectively). Conclusions: These data suggest a complementary role of MRI to CT to reduce the risk of geographical misses, although they highlight the potential for larger target volumes and hence toxicity.

## 1. Introduction

Intensity modulated radiotherapy (IMRT) provides steep dose gradients, facilitating lower doses to adjacent structures, but also has potential for geographical misses if tissues containing tumour are not accurately encompassed within the target volume [1,2,3]. However, the delineation of primary tumours is associated with considerable uncertainty [4,5], and variation in this process can have a major impact upon dose to primary tumour and to organs at risk (OAR) [6].

Recent international consensus guidelines for delineation of laryngeal, hypopharyngeal, oropharyngeal, and oral cavity primary tumour clinical target volumes (CTV) have recommended a volumetric approach based on the Danish Head and Neck Cancer Group (DAHANCA) approach of a ‘5 + 5 mm’ expansion margin (5 mm to high dose and a further 5 mm to intermediate or prophylactic dose) from gross tumour volume (GTV), with subsequent anatomical modification based upon tumour site and T stage [7]. This volumetric approach, with limited margins to CTV, places considerable importance upon accurate GTV delineation. As highlighted within these guidelines, the integration of clinical examination/endoscopy is a vital step in accurate GTV delineation; however, the use of the planning CT remains fundamental to primary tumour delineation.

CT imaging provides geometric accuracy and accurate dosimetric data based on tissue electron density, which is calibrated from Hounsfield units [8]. However, accurate delineation of an oropharyngeal tumour can be very challenging on CT, with limited soft tissue contrast and susceptibility to dental artefacts [9]. Consequently, CT-based delineation of oropharyngeal tumours is associated with considerable inter-observer [9,10] and intra-observer variability [11].

Integrating multi-modality imaging into the radiotherapy planning process provides an opportunity to improve upon delineation directly on CT, with side-by-side assessment of diagnostic imaging. Compared with CT, MRI provides superior soft tissue contrast and less susceptibility to dental artefacts [12], and diagnostic MRI is considered superior for visualising oropharyngeal tumours [7,9,12]. The use of MRI for delineation has been shown to reduce both inter-observer [10,13,14] and intra-observer variability [11].

A ‘gold-standard’ for incorporating MRI directly into the radiotherapy planning pathway is by acquisition of a dedicated MRI for planning in the radiotherapy planning position, using the same immobilisation devices, with subsequent rigid registration to the planning CT [13,15,16]. However, the majority of radiotherapy departments do not have access to a dedicated MRI scanner for radiotherapy planning. An alternative is registration of a diagnostic MRI scan, acquired optimally for radiotherapy purposes, to the planning CT scan. In a planning study, we have previously shown that deformable registration of a diagnostic MRI offers a significant advantage in the accuracy of GTV delineation compared with planning CT alone [11].

In the absence of a dedicated MRI acquired for radiotherapy planning, we have implemented the pathway of using deformable registration of a diagnostic MRI, acquired optimally for radiotherapy purposes, to planning CT into routine clinical practice. The purpose of this report is to assess the impact of the integration of a registered MRI upon GTV delineation of oropharyngeal carcinoma in routine clinical practice.

## 2. Methods

### 2.1. Study Population

This single centre retrospective study was approved by the institutional review board. Inclusion criteria for the study were: (1) histologically proven primary oropharyngeal squamous cell carcinomas (SCC); (2) treated with definitive radiotherapy +/− chemotherapy; (3) primary tumour identifiable on diagnostic MRI; (4) diagnostic MRI optimised for radiotherapy deformably registered to planning CT scan for delineation of the primary tumour; (5) separate GTVs delineated in routine clinical practice using CT (GTV_CT_), MRI (GTV_MR_), and CT and MRI (GTV_CTMR_); (6) all patients outlined by the same radiation oncologist; and (7) planning CT between February 2015 and December 2016.

### 2.2. Image Acquisition

#### 2.2.1. MRI

Diagnostic MRI images were obtained on a 1.5T Siemens Magnetom Avanto (Siemens Healthcare, Erlangen, Germany) using axial T1-weighted sequences with a TR = 831 ms, TE = 8.6 ms, 105 2-mm thick contiguous slices and a voxel size = 0.9 × 0.9 × 2.0 mm. The sequence was a T1W fast spin echo sequence, with a number of scan averages of 2; the echo train length was 2 and parallel imaging using the GRAPPA method was used with a parallel imaging factor of 2. The patient was positioned on a standard curved diagnostic couch without an immobilisation mask. The use of a 2 mm contiguous slice thickness on MRI was a change to standard diagnostic practice in order to optimise the image registration to planning CT.

#### 2.2.2. CT

Planning CT (pCT) images were optimised for use in treatment planning. CT imaging was performed on a Siemens Somatom Sensation 64-section (Siemens Healthcare, Erlangen, Germany). Images were acquired with the patient on a flat top radiotherapy couch in a five-point thermoplastic radiotherapy immobilization mask, and were contrast enhanced after a 25-s delay following a bolus of 100 mL of iodinated contrast (Omnipaque 300, Bracco Ltd., High Wycombe, UK) injected at 3 mL/s using the following settings: 120 kV, variable mA (min 10, max 600, noise index 12.2), tube rotation 1.0 s per rotation, pitch 0.9, with a 2.0 mm section reconstruction. The voxel size used was 1.0 × 1.0 × 2.0 mm.

### 2.3. Deformable MRI–CT Image Registration

The T1 weighted MRI and pCT images were registered using deformable image registration (DIR). All registrations were performed with Mirada RTx v1.4 (Mirada Medical, Oxford, UK) by a trained radiotherapy dosimetrist. An automatic rigid fast algorithm was used for initial rigid registrations, and as the starting point for the DIR. If the automatic rigid registration results were not deemed acceptable, then a manual rigid registration was performed. DIR was then performed based upon a multi-modal, mutual information-based algorithm. All registrations were checked by the treating radiation oncologist by visual interrogation of the fused images; this involved assessing overlay images of regions of anatomy e.g., spinal cord, bone, and mucosa-air interfaces in proximity to the GTV.

### 2.4. GTV Delineation

Three contours per patient were produced by a single radiation oncologist specialising in head and neck cancer on Monaco v 5.1 (Elekta AB, Stockholm, Sweden). All contours were delineated viewing all available diagnostic imaging (including MRI) side-by-side. For each patient, three GTV contours had been manually delineated as part of routine clinical practice, according to an institutional protocol in the following order during the same session: GTV on the CT only without registration (GTV_CT_), GTV on the T1-MRI only (GTV_MR_) without viewing registered planning CT, and a final GTV based on viewing the GTV_CT_ and GTV_MR_ and registered images (GTV_CTMR_).

### 2.5. Analysis of GTVs

The GTVs were quantitatively analysed by overall volume, and positional metrics using five contour comparison metrics calculated using ImSimQA v3.1 (OSL, Shrewsbury, UK). Three sets of comparisons were made: (i) GTV_CTMR_ (reference) vs GTV_CT_, (ii) GTV_CTMR_ (reference) vs. GTV_MR_, and (iii) GTV_CT_ (reference) vs. GTV_MR_. The five contour comparison metrics used are described below.

#### 2.5.1. Conformity Index (CI)

CI is the ratio of the area of the overlapping region to the total area of the contours [17]. A value between 0 and 1 is given. 0 shows no overlap between the two GTVs. 1 shows identical GTVs.
(1)$$\mathrm{CI}=\frac{B}{A+B+C}$$
*B* represents the volume of the overlapping regions. *A* and *C* represent the volume of each GTV which is not overlapping.

#### 2.5.2. Mean Distance to Conformity (MDC)

MDC is mean distance of each outlying voxel of the evaluation contour surface to the closest voxel from the reference contour surface [17]. MDC quantitates global mismatch in the shape between the two contours. MDC for a perfect overlap would be 0 mm.

#### 2.5.3. Dice Similarity Coefficient (DSC)

DSC also quantifies the overlap between the two contours, with 1 showing complete overlap and 0 showing no overlap [17,18].
(2)$$\mathrm{DSC}=\frac{2\cdot \left(A\cap B\right)}{A+B}$$
where *A* is the volume of the reference contour and *B* is the volume of the evaluation contour.

#### 2.5.4. Centre of Gravity Distance (CGD)

CGD is the distance between the geometrical centres of the two GTVs being compared. CGD for a perfect match would be 0 mm [17].

#### 2.5.5. Volumetric Analysis

The volumes were compared by looking at a percentage difference between the volumes of the contours. Perfectly correlating contours would have a percentage difference of 0.

### 2.6. Statistical Analysis

Wilcoxon signed rank tests were performed on StataSE 13 (StataCorp, College Station, TX, USA) to test for significant difference in volumes between (i) GTV_CT_ and GTV_MR_, (ii) GTV_CTMR_, and GTV_CT_, and (iii) GTV_CTMR_ and GTV_MR_. Results were considered statistically significant if the *p*-value was <0.02. A *p*-value of 0.02 was used to minimise the risk of Type I errors arising from multiple testing.

## 3. Results

22 consecutive patients who underwent radiotherapy for oropharyngeal carcinoma, and for whom primary tumour GTVs were contoured by a single radiation oncologist, were identified. A total of 9/22 (41%) and 10/22 (45%) patients had a tonsil and base of tongue primary respectively. Contouring was performed with the aid of deformable registration of a diagnostic MRI to the planning CT in routine clinical practice between February 2015 and December 2016. Patient and tumour characteristics are summarised in Table 1. All tumours were non-metastatic. The median time from diagnostic MRI to planning CT was 22 days (range 0–50). All patients had a diagnostic PET-CT, which was available for side-by-side assessment (not registered to planning CT) at the time of delineation. Figure 1 provides three examples of patients with small, medium, and large primary tumours, for whom deformable image registration of the MRI to the planning CT was undertaken, and the GTV_CT_, GTV_MR_, and GTV_CTMR_ which were contoured as part of routine clinical practice; GTV_CTMR_ was used as the basis for delineation of subsequent clinical target volumes (CTV) and planning target volumes (PTV) for treatment planning.

### 3.1. Comparison of GTV Volumes

The volumes of GTV_CT_, GTV_MR_ and GTV_CTMR_ for the 22 patients detailed in Table 2 are summarised in Figure 2. GTV_CTMR_ was the largest volume for all 22 patients. Median GTV volumes were 13.7 cm^3^ (range 3.5–41.7), 13.4 cm^3^ (range 1.6–38.3), and 17.8 cm^3^ (range 5.5–44.5) for GTV_CT_, GTV_MR_, and GTV_CTMR_ respectively. As shown in Table 2, there was no significant difference in GTV_CT_ and GTV_MR_ volumes. GTV_CTMR_ was found to be significantly larger than both GTV_MR_ and GTV_CT_. GTVs for tonsil primary lesions were significantly higher than for non-tonsil primary lesions. Mean and median GTV_CTMR_ for tonsil primary sites were 28.2 cm^3^ and 30.8 cm^3^, versus 13.7 cm^3^ and 11.6 cm^3^, respectively, for non-tonsil primary sites (*p* = 0.01).

### 3.2. Comparison of Positional Metrics

The analysis of positional contour variability is summarised in Table 3. Based on all four positional metrics, GTV_CT_ and GTV_MR_ were the least similar (for example, mean DSC 0.71, 0.84, 0.82 for GTV_CT_–GTV_MR_, GTV_CTMR_–GTV_CT_ and GTV_CTMR_–GTV_MR_, respectively). Positional metrics (DSC, conformality index, MDC and CGD) comparing GTV_CT_–GTV_MR_ were significantly different when compared with those derived from either GTV_CTMR_–GTV_CT_ or GTV_CTMR_–GTV_MR_ analysis. By contrast, there were no significant differences in positional metrics obtained from comparing CT_CTMR_–GTV_MR_ and those comparing CT_CTMR_–GTV_CT_ (Table 3). There were no significant differences in DSC, conformity index, MDC, or CGD comparing GTV_CT_–GTV_MR_, GTV_CTMR_–GTV_CT_ and GTV_CTMR_–GTV_MR_ for tonsil versus non-tonsil primaries.

## 4. Discussion

Accurate GTV delineation is a critical issue in head and neck radiotherapy, with the use of IMRT with steep dose gradients [12] and the introduction of new guidelines with tight volumetric and anatomical margins to CTVs [7]. Target volume delineation remains a major source of variability in the planning process [6,19,20]. Recurrence analyses have shown that geographical misses remain a key issue in the era of IMRT; in an analysis of locoregional recurrences referred to a tertiary centre, 41% were classified as marginal, [3] implying inadequate GTV delineation. The integration of multimodality imaging into the radiotherapy planning process holds the potential to considerably improve the accuracy and reproducibility of GTV delineation. Delineation of oropharyngeal carcinomas is particularly challenging on CT, with very limited soft tissue resolution and problems with dental artefacts [9,12], with MRI offering superiority in diagnostic pathways [7]. MRI-based delineation is widely considered to be more reproducible [10,11,21]. Although MRI is associated with geometric spatial distortion, this is very small for a limited cross-sectional area such as the head and neck region [13]. However, MRI does not provide the electron density information required for dosimetric calculation, and current clinical use requires registration to the planning CT.

Registration of an MRI acquired in treatment position to planning CT is considered the preferred method for the use of MRI in treatment planning. In common with many centres, we do not have dedicated access to an MRI acquired for planning purposes. In a prospective planning study in 14 patients [11] (of whom 11 had an oropharyngeal carcinoma), we defined a GTV based on a dedicated MRI acquired in the treatment position as a reference standard; deformable registration of a diagnostic MRI to planning CT led to GTV delineation which was significantly closer than based on planning CT alone (with side-by-side assessment of diagnostic imaging). Based upon these findings we have introduced deformable registration of a diagnostic T1-weighted MRI acquired at 2 mm contiguous slice thickness to the planning CT into clinical practice. This analysis reports on the impact of this pathway upon delineation of a final GTV used for planning in the reality of clinical practice.

In the retrospective analysis, all GTVs were contoured as part of routine clinical practice. There was no overall significant difference in the volume of primary tumours delineated using CT or MR alone. Consistent with this data, a previous planning study of 11 patients with oropharyngeal carcinoma found no significant difference in overall volumes outlined by multiple observers comparing GTVs delineated on CT or MRI alone, although GTV delineation on MRI was associated with a significantly smaller volume standard deviation than on CT [10]. Data are variable in other planning studies across differing head and neck tumour sites comparing CT- and MRI-defined GTVs, with some showing similar overall GTVs [22,23,24], although some studies have suggested MRI-defined GTVs are smaller [25,26] or larger [9,27,28]. In this current study, GTVs delineated using both MRI and CT (GTV_CTMR_) were significantly larger than those delineated using CT or MRI alone. Our prior planning study also showed that GTVs based on CT were significantly smaller than those using combined CT–MRI, although there were non-significant differences between GTVs based on MRI alone and combined CT–MRI [10].

Positional metric analysis demonstrated only limited similarity between GTV_CT_ and GTV_MR_ (DSC 0.71), despite the availability of diagnostic MRI (and PET-CT in 100% of patients) to guide delineation. There was more similarity between GTV_CTMR_ and both GTV_CT_ and GTV_MR_ (DSC 0.84 and 0.82 respectively). Positional metrics comparing GTV_CT_–GTV_MR_ were significantly different when compared with those derived from either GTV_CTMR_–GTV_CT_ or GTV_CTMR_–GTV_MR_ analysis. There was no significant difference in positional metric analysis between tonsil and non-tonsil primary tumours.

The findings of the volume and positional analysis suggests that the radiation oncologist used both GTV_CT_ and GTV_MR_ in determining the final GTV_CTMR_, which was used as the basis for subsequent CTV and PTV delineation according to institutional protocols. Therefore, both CT and MRI appear to be potentially complimentary in GTV delineation. Overall, clinical implementation of deformable registration of diagnostic MRI into clinical practice led to delineation of significantly larger GTVs than those delineated on CT alone. Dental artefacts and lack of soft tissue contrast on CT provide a clear rationale to integrate MRI directly into the planning process. Based upon this premise, these data suggest that this pathway may reduce the risk of geometric misses by more accurately identifying the GTV. However, conversely, multimodality imaging which leads to larger GTVs will increase high dose planning target volumes (PTV) (assuming the same protocols are used in generating CTVs and PTVs); this has the potential to increase treatment-related toxicity.

Limitations of this study include the use of a T1-weighted MRI without gadolinium; although gadolinium may assist in delineation, we do not have a suitable diagnostic MRI sequence currently available with a slice thickness that facilitates accurate registration (due to increased scan time required and constraints on MRI availability). However, enhanced sequences are available for side-by-side comparison with the registered non-enhanced T1-sequence. The analysis is based upon GTV delineation by a single experienced observer who had received instruction via head and neck radiologists in the use of MRI. Inter-observer variation in GTV delineation is well recognised [9,10,20], and is likely to have been seen if further observers had been included. However, the overall results are consistent with those of our prior multi-observer planning study in oropharyngeal carcinoma [10]. The diagnostic MRI was acquired prior to the planning CT (median interval 22 days); there is potential for tumour growth in the intervening period. The underlying premise for the integration of MRI into the planning process is that this will enhance the accuracy of GTV delineation; however, there are no correlative pathological data available as a ‘ground truth’ for oropharyngeal carcinoma to evaluate the accuracy of GTVs constructed using CT alone, MRI alone or a combination of both.

## 5. Conclusions

This study of clinical practice shows that the use of deformable registration of diagnostic MRI to planning CT for oropharyngeal carcinomas produced significantly larger primary tumour GTVs compared to the use of CT alone, with significant positional differences. These data suggest a complementary role of MRI in the radiotherapy planning process for oropharyngeal carcinoma to reduce the risk of geographical misses. This is potentially of increased importance in light of recent guidelines of tighter GTV to CTV margins. These data also show that combined modality imaging is likely to lead to larger target volumes and consequently potential for increased toxicity.

## Figures and Tables

**Figure 1 healthcare-06-00135-f001:**
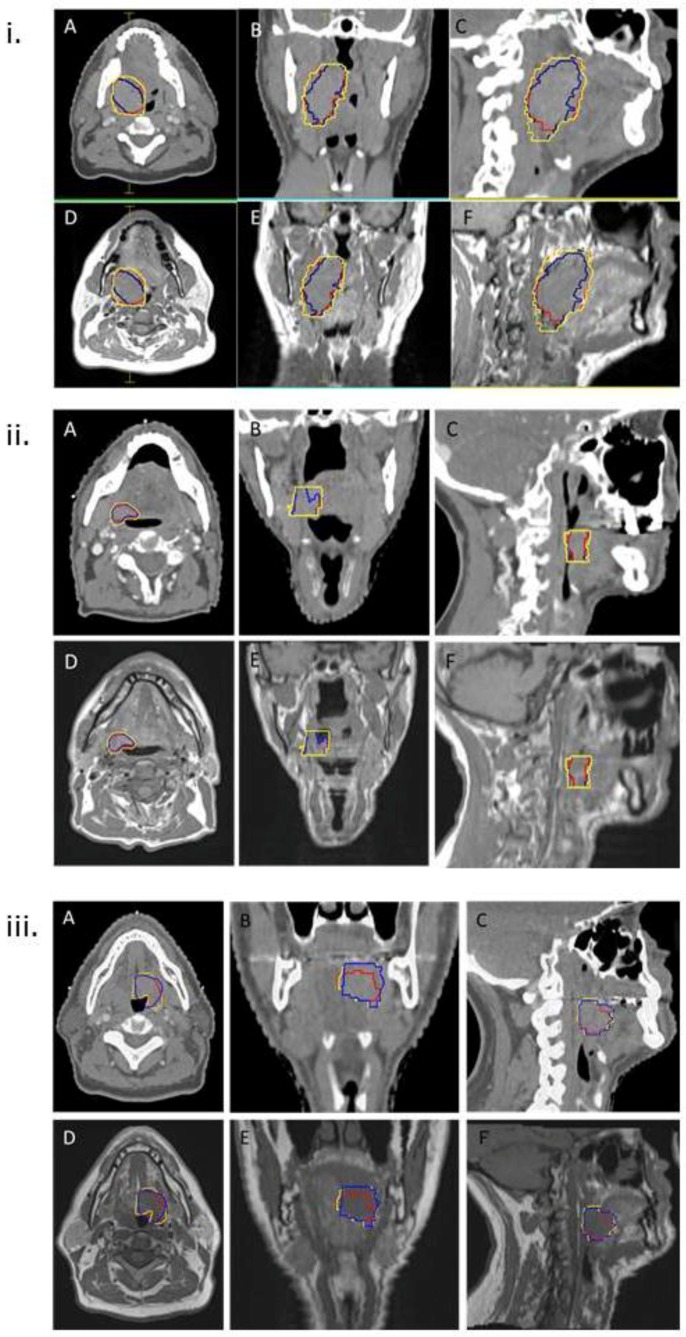
Examples of delineation of primary tumour GTV using CT and deformably coregistered MRI in patients with different size primary tumours. (i) Example of large primary right tonsil carcinoma T3N2bM0 (Patient 1 in Table 1). (ii) Example of small primary right base of tongue carcinoma T1N2bM0 (Patient 20 in Table 1). (iii) Example of medium sized primary base of tongue carcinoma T4aN2bM0 (Patient 6 in Table 1). GTV_CT_ (red), GTV_MR_ (blue) and GTV_CTMR_ (yellow) are shown. For each patient in (i), (ii), and (iii) axial, coronal, and sagittal images of planning CT are shown in (**A**–**C**) respectively; (**D**–**F**) show axial, coronal, and sagittal images of deformably coregistered T1-weighted diagnostic MRI respectively.

**Figure 2 healthcare-06-00135-f002:**
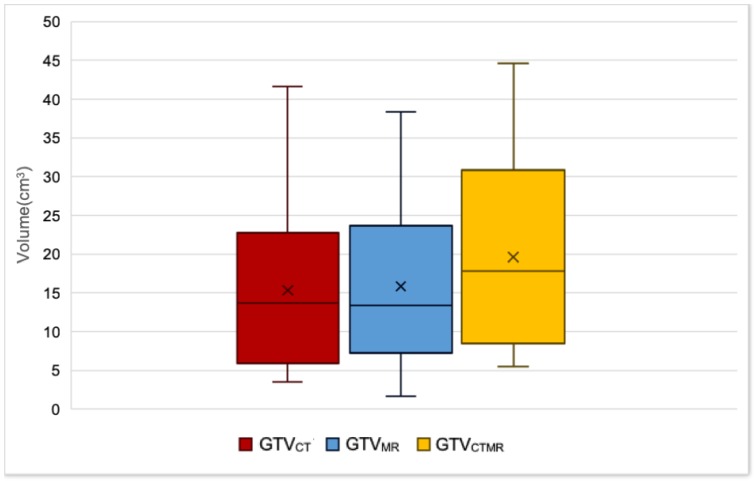
Box plot summarising volumes of GTV_CT_ (red), GTV_MR_ (blue) and GTV_CTMR_ (yellow). The x denotes the mean, and the horizontal line in each box the median value.

**Table 1 healthcare-06-00135-t001:** Table to show patient demographics and tumour characteristics.

Patient Number	Sex	Age (Years)	Subsite	T-Stage	N-Stage	Grade	p16 Status	GTV_CT_ (cm^3^)	GTV_MR_ (cm^3^)	GTV_CTMR_ (cm^3^)
1	M	52	Tonsil	3	2b	3	positive	39.1	32.5	44.6
2	M	56	Tonsil	2	2b	3	Positive	4.7	1.6	6.3
3	M	45	Base of tongue	4a	2b	3	Positive	6.1	8.7	10.1
4	M	61	Base of tongue	2	2b	3	Positive	5.3	5.8	7.0
5	M	53	Tonsil	4b	3	3	Positive	20.2	21.9	27.2
6	M	59	Base of tongue	4	2b	3	Positive	12.5	15.3	17.8
7	M	57	Tonsil	3	3	3	Positive	17.7	15.7	23.3
8	F	53	Base of tongue	2	2b	3	Positive	6.4	7.1	8.5
9	F	66	Pharyngeal wall	3	1	3	Positive	21.2	18.5	23.0
10	F	50	Base of tongue	2	1	3	Negative	7.4	7.2	8.1
11	M	55	Tonsil	4a	2b	2	Positive	22.3	29.1	30.8
12	F	61	Base of tongue	4a	2b	3	Unknown	24.6	28.1	31.0
13	M	53	Tonsil	2	2b	3	Positive	14.8	9.6	17.7
14	M	65	Base of tongue	2	2c	3	Positive	4.7	11.0	11.6
15	M	58	Tonsil	4a	2b	3	Negative	24.0	26.5	30.8
16	M	49	Tonsil	3	0	3	Positive	24.9	22.7	30.8
17	M	56	Base of tongue	4a	2b	3	Positive	17.0	18.8	22.7
18	F	62	Pharyngeal wall	1	2b	3	Unknown	3.5	6.0	6.3
19	F	62	Base of tongue	2	2	3	Negative	7.1	9.0	11.8
20	M	53	Base of tongue	1	2b	2	Unknown	3.8	3.9	5.5
21	M	66	Tonsil	2	2b	3	Unknown	41.7	38.3	41.9
22	F	66	Soft palate	4a	1	3	Unknown	8.3	11.5	14.7

**Table 2 healthcare-06-00135-t002:** Summary of volume of GTVs contoured using CT, MR, and deformably registered MRI to CT. Statistically significant differences in bold (*p* < 0.02).

Modality	Modality GTV Volumes (cm^3^)					
	Mean	Median	Mean St. Dev.	Range		GTV Comparison (Wilcoxon Signed Rank Test)
				Max	Min	
CT	15.3	13.7	11.1	41.7	3.5	GTV_CT_–GTV_MR_, *p* = 0.47
MR	15.9	13.4	10.2	38.3	1.6	GTV_CTMR_–GTV_CT_, *p* = 0.018
CT–MR	19.6	17.8	11.8	44.6	5.5	GTV_CTMR_–GTV_MR_, *p* = 0.010

**Table 3 healthcare-06-00135-t003:** Intermodality positional GTV analysis.

Metric		DSC	Conformity Index	MDC (mm)	CGD (mm)
GTV_CTMR_−GTV_CT_ Mean (SD)		0.84 (0.10)	0.66 (0.12)	3.74 (1.19)	2.02 (1.40)
GTV_CTMR_−GTV_MR_ Mean (SD)		0.82 (0.10)	0.65 (0.14)	4.03 (1.10)	1.86 (1.59)
GTV_CT_−GTV_MR_ Mean (SD)		0.71 (0.13)	0.50 (0.15)	5.09 (1.89)	3.08 (2.37)
Statistical Comparisons of Positional Metrics: *p* Values	GTV_CT_−GTV_MR_ versus GTV_CTMR_−GTV_CT_	<0.01	<0.01	<0.01	<0.01
	GTV_CT_−GTV_MR_ versus GTV_CTMR_−GTV_MR_	<0.01	<0.01	<0.01	<0.01
	GTV_CTMR_−GTV_MR_ versus GTV_CTMR_−GTV_CT_	0.71	0.82	0.15	0.70

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
