# Peer review of "Assessment of the Impact of Deformable Registration of Diagnostic MRI to Planning CT on GTV Delineation for Radiotherapy for Oropharyngeal Carcinoma in Routine Clinical Practice"

_healthcare, 2018, doi:10.3390/healthcare6040135_

Round 1

Reviewer 1 Report

Dear Authors,

Congratulations on successfully presenting the work titled 'Assessment of the Impact of Deformable Registration of Diagnostic MRI to Planning CT on GTV Delineation for Radiotherapy for Oropharyngeal Carcinoma in Routine Clinical Practice'. The manuscript is well structured and presented with appropriate methodology and results. The conclusions are also well supported by results.

I have only one concern. That is, in Figure 1, the regions being studied and imaged could be mentioned on the images.

Also, the type of cancer being investigated could also be labelled on the images in Figure 1.  

Author Response

We would like to thank the Peer Reviewers for their helpful input. Our responses and summaries of modifications made to the manuscript are detailed below in bold.

Congratulations on successfully presenting the work titled 'Assessment of the Impact of Deformable Registration of Diagnostic MRI to Planning CT on GTV Delineation for Radiotherapy for Oropharyngeal Carcinoma in Routine Clinical Practice'. The manuscript is well structured and presented with appropriate methodology and results. The conclusions are also well supported by results.

I have only one concern. That is, in Figure 1, the regions being studied and imaged could be mentioned on the images.

Also, the type of cancer being investigated could also be labelled on the images in Figure 1.

The caption has been amended to include details of the tumour, patient number corresponding to Table 1 and the type of imaging shown corresponding to labels added to each sub-part of the figure A-F. In line with suggestions from Reviewer 2 further examples of small and medium sized tumors have been added to this Figure.

Reviewer 2 Report

In this manuscript by Taylor et al, the authors shown the complimentary role MRI into the radiotherapy planning process for oropharyngeal carcinoma. I have some suggestions/concerns about this manuscript.

Major:

1.       Table 2: It shows for MT both Mean and Median are 15.9 same is true for CT-MR; I don’t think the calculation is right-PLEASE CHECK; Please also recheck the Wilcoxin test for GTV(CT) vs GTV(MR)

2.       The way Figure 2 is presented, it is redundant (in presence of Table 1). It would be more informative if the author can present the Figure 2 as scatter dot blot distinguishing three different measures (CT, MR ant CT-MR); the difference of CT-MR with other two measure will then be more appreciable from the plot. Mean median values for each measures can also be presented in the plot.

3.       Planning CT and non-contrast T1 MR are shown for only one patient (with high values for all the 3 parameters). It would be prudent to show the similar pictures for three representative patients, with high (e.g. Patient#1, already presented) low (e.g. patient #2 or 20) and median (e.g. patient # 9 or 17) values of GTVs.

4.       It is not clear whether the different matrices presented in table 3 are significantly different among the values presented in each column.

5.       Is there any significant difference in GTV values in patients with tonsillar cancer vs patients having cancer in other part of the oropharynx ( e.g. base of tongue)?

Minor:

6.       Minor: Line 2: “were”-repeated

7.       Table 2. Max value for CT-MR (as per Table 1) is 44.6, not 44.5

Author Response

We would like to thank the Peer Reviewers for their helpful input. Our responses and summaries of modifications made to the manuscript are detailed below in bold.

In this manuscript by Taylor et al, the authors shown the complimentary role MRI into the radiotherapy planning process for oropharyngeal carcinoma. I have some suggestions/concerns about this manuscript.

Major:

1.       Table 2: It shows for MT both Mean and Median are 15.9 same is true for CT-MR; I don’t think the calculation is right-PLEASE CHECK; Please also recheck the Wilcoxin test for GTV(CT) vs GTV(MR)

Thank you. The median values had been incorrectly transcribed and have been corrected.  The Wilcoxon test has been checked and is correct.

2.       The way Figure 2 is presented, it is redundant (in presence of Table 1). It would be more informative if the author can present the Figure 2 as scatter dot blot distinguishing three different measures (CT, MR ant CT-MR); the difference of CT-MR with other two measure will then be more appreciable from the plot. Mean median values for each measures can also be presented in the plot.

The bar chart previously in figure two has been changed to a more informative way to present the data as suggested. A box plot has been used as it provides a clear visual summary of the data and displays the mean and median.

3.       Planning CT and non-contrast T1 MR are shown for only one patient (with high values for all the 3 parameters). It would be prudent to show the similar pictures for three representative patients, with high (e.g. Patient#1, already presented) low (e.g. patient #2 or 20) and median (e.g. patient # 9 or 17) values of GTVs.

As suggested we have added images of CT and MRI in axial/coronal/sagittal planes for 2 further patients, one with a small and one with a medium sized primary tumour to provide more representative images for the reader.

4.       It is not clear whether the different matrices presented in table 3 are significantly different among the values presented in each column.

Statistics of comparison between the positional metrics have been added to the table to compare the columns already presented.  This has also been highlighted in the Results section ‘Comparison of Positional Metrics’ with the lines added:

‘Positional metrics (DSC, conformality index, MDC and CGD) comparing GTVCT-GTVMR were significantly different when compared with those derived from either GTVCTMR-GTVCT or GTVCTMR-GTVMR analysis.  By contrast there were no significant differences in positional metrics obtained from comparing CTCTMR-GTVMR and those comparing CTCTMR-GTVCT (Table 3).’

Reference made to this analysis also added to Paragraph 4 of Discussion.

5.       Is there any significant difference in GTV values in patients with tonsillar cancer vs patients having cancer in other part of the oropharynx ( e.g. base of tongue)?

In this limited size sample GTV volumes were statistically larger than non-tonsillar volumes. An additional line has been added to the start of the Results section: ‘9/22 (41%) and 10/22 (45%) patients had a tonsil and base of tongue primary respectively.’ In the section of the results entitled ‘Comparison of GTV volumes’ the following lines have been added:

‘GTVs for tonsil primary lesions were significantly higher than for non-tonsil primary lesions.  Mean and median GTVCTMR for tonsil primary were 28.2 cm3 and 30.8 cm3 versus 13.7 cm3 and 11.6 cm3 respectively for non-tonsil primary sites (p=0.01).’

In addition, the following lines have been added to ‘Comparison of positional metrics’ concerning data re. tonsil v non-tonsil:

‘There were no significant differences in DSC, conformity index, MDC or CGD comparing GTVCT-GTVMR, GTVCTMR-GTVCT and GTVCTMR-GTVMR for tonsil versus non-tonsil primaries.’

These data comparing positional metrics show non-significant differences and is based on a small subset analysis so we did not think it would be helpful to show this data as a separate table but rather to summarise in this manner.  This comparison is also now added to paragraph 4 of Discussion.

Minor:

6.       Minor: Line 2: “were”-repeated

Corrected.

7.       Table 2. Max value for CT-MR (as per Table 1) is 44.6, not 44.5

Corrected.

Round 2

Reviewer 2 Report

The authors have addressed my concerns sufficiently and satisfactorily to recommend publication of the manuscript in its current form.